# INTEGRATE: Distance based Graph Convolutional Networks for Statistical Relational Learning

## Abstract

Recently, several successful methods for learning embeddings of large knowledge bases have been developed. They have been motivated through the necessity of learning and reasoning about various entities, their attributes and relations present in the knowledge bases. A potential limitation of much of this line of research is that the inherent multi-relational structure of the network is not exploited since only the node features are taken into account by these set of methods. To overcome this limitation, graph convolutional networks (GCNs) were proposed that generalized neural network models to multi-relational, graph-structured data sets. We consider the problem of learning distance-based Graph Convolutional Networks (GCNs) for multi-relational data within statistical relational learning. Specifically, we first embed the original graph into the Euclidean space $\mathbb{R}^m$ using a relational density estimation technique thereby constructing a secondary Euclidean graph. The graph vertices correspond to the target triples and edges denote the Euclidean distances between the target triples. We emphasize the importance of learning the secondary Euclidean graph and the advantages of employing a distance matrix over the typically used adjacency matrix. Our comprehensive empirical evaluation demonstrates the superiority of our approach over 18 approaches spread over different GCN models, relational embedding techniques, rule learning techniques and relational models.

## 1 Introduction

Recently, several successful methods for learning embeddings of large knowledge bases have been developed which have been motivated through the necessity of learning and reasoning about various entities, their attributes and relations present in the knowledge bases (Lin et al., 2015; Shi & Weninger, 2017; Wang et al., 2017). A potential limitation of much of this line of research is that the inherent multi-relational structure of the network is not exploited since only the node features are taken into account by these set of methods. To overcome this limitation, graph convolutional networks (GCNs) (Defferrard et al., 2016; Kipf & Welling, 2017) were proposed that generalized neural network models to multi-relational, graph-structured data sets. Similar to the convolution operators in convolutional neural networks (CNNs) that extract locally stationary features in the inputs data, GCNs utilize the graph convolution operator defined with respect to the adjacency matrix to extract local features from a semantic point of view. The main reason behind the success of GCNs is that they exploit two key types of information: node feature descriptions and node neighborhood structure (captured through the adjacency matrix of the graph). While successful, GCNs still have a limitation in that they cannot directly be applied on multi-relational data/networks.

Statistical Relational Learning (SRL) (Koller et al., 2007; Raedt et al., 2016) combines the power of probabilistic models to handle uncertainty with the ability of relational models to faithfully capture the rich domain structure. One of the key successes of these models lie in the task of knowledge base population, specifically, link prediction and node classification. While successful, most methods make several simplifying assumptions – presence of supervision in the form of labels, closed-world assumption, presence of only binary relations and most importantly, in many cases, presence of hand-crafted domain rules.

We go beyond these assumptions and inspired by this recent success of Graph Convolutional Networks (GCNs) develop a *new framework for relational GCNs* for statistical relational learning. This framework has two key steps: (1) create a secondary Euclidean graph from the original graph by *learning* rules from one-class data, i.e., from the positive and negative annotations of the target relation separately. The next step is to *convert* these rules into observed features i.e., instantiate and count the number of times the rules fire and computes the distance matrix, and (2) finally, it *trains* a GCN using the observed features and the distance matrix. For the first step, our method employs a one-class density estimation method that employs a tree-based distance metric to learn relational rules iteratively. Hence, we call the framework as *relat**I**onal de**N**sity dis**T**ance bas**E**d **GRA**ph convolu**T**ional n**E**tworks* (INTEGRATE). Since the two different steps of learning the relational rules and training the GCN employ the same set of positive examples, a richer representation of the combination of the attributes, entities and their relations is obtained. While previous methods used the features as the observed layer, INTEGRATE uses the rules as the observed layer. This has the added advantage of the latent layer being richer – it combines the instantiations of first-order rules themselves allowing for a richer representation. We hypothesize and show empirically that this is specifically useful when employed on link prediction and node classification tasks. Although work exists on generating similarity graphs using GNNs (Bai et al., 2018; 2019; Li et al., 2019), ours is the first method to use GCNs on induced similarities graphs allowing for use of richer features.

Taking advantage of the graph structure and learning first-order rules can have a large impact on various real world applications such as drug discovery, traffic prediction and real-world physical systems to name a few. This is due to two reasons: (i) Features constructed by our method encapsulate information about entities as well as the relationship between the entities in the relational space, and (ii) expert knowledge can be explicitly encoded in the rules by either learning from this expert knowledge or by modifying the learned rules.

We make the following key contributions:

(1) We develop the first relational GCN capable of utilizing the different densities of the data separately.

(2) Going beyond using carefully designed hand-crafted rules, our method learns rules automatically to construct a secondary graph and constructs the GCN. These two steps are conditioned on the required task and allow for a better classifier and thus can learn with *smaller data*.

(3) INTEGRATE can handle arbitrary relations – not simple binary relations that most methods use. Given that our base learner employs a logic learner, the relations can be $n-$ary.

(4) We show the advantages of using distance matrices and Euclidean distance to construct the distance matrix.

Our evaluation across 14 different baselines and 7 different data sets clearly demonstrates the effectiveness of INTEGRATE. The rest of the paper is structured as follows: We first start with introducing the necessary background before introducing the building blocks of INTEGRATE, showing its effectiveness by extensive experimental evaluation before concluding.

## 2   Background and Related Work

**Notations:** A (logical) **predicate** is of the form $\mathcal{R}(t_1, \ldots, t_k)$ where $\mathcal{R}$ is a relation and the arguments $t_i$ are **entities**. A **substitution** is of the form $\boldsymbol{\theta} = \{\langle l_1, \ldots, l_k \rangle / \langle t_1, \ldots, t_k \rangle\}$ where $l_i$s are logical variables and $t_i$s are terms. A **grounding** of a predicate with variables $l_1, \ldots, l_k$ is a substitution $\{\langle l_1, \ldots, l_k \rangle / \langle L_1, \ldots, L_k \rangle\}^1$ mapping each of its variables to a constant in the domain of that variable. A knowledge base $\mathcal{B}$ consists of (1) entities: a finite domain of objects $\mathcal{O}$, (2) relations: a set of predicates describing the attributes and relationships between objects $\in \mathcal{O}$, and (3) an interpretation assigning a truth value to every grounding of a predicate.

**Relational Density Estimation:** A common issue in many real-world relational knowledge bases is that only true instances of any relation(s) are labeled while the false instances are not explicitly identified. Consequently

---

[1]We use uppercase for relations/groundings and lowercase for variables.

*closed-world assumption* is applied to sample negative instances. While reasonable, this is a strong assumption particularly when the number of positively labeled examples $\ll$ negatively labeled examples. In the relational one-class classification (Khot et al., 2014) method, given a set of labeled examples, a distance measure is used to perform one-class classification, which involves two levels of combinations: tree-level due to learning multiple trees and instance-level due to the predicates containing variables and different instances for each target. For example, in learning *advisedBy(S,P)* the first tree could consider the courses and the second could consider the publications. The tree-level combining function combines the results from these two trees. Now the student could potentially publish several papers, or register in multiple courses and inside each tree, these different instances are combined using the instance level combining function (Jaeger, 2007; Natarajan et al., 2008). In the tree-level distance computation, the distance between the current unlabeled example $u$ is calculated from a labeled example in all the learned first-order trees. Now the final distance is simply the weighted combination of the individual tree-level distances: $D(l_1, u) = \sum_i \beta_i \, d_i(l_1, u)$ where $\beta_i$ is the weight of the $i^{th}$ tree and $\sum_i \beta_i = 1, \beta_i \geq 0$. These tree distances are then combined to get an overall distance between the current example and all the labeled examples $l_j$, $E(u \notin \texttt{class}) = \sum_j \alpha_j D(l_j, u)$, where $\alpha_j$ is the weight of the labeled example $l_j$ and $\sum \alpha_j = 1, \alpha_j \geq 0$.

**Knowledge Graph Embeddings (KGEs):** Recently, several successful methods for learning embeddings of large knowledge bases have been developed (Wang et al., 2017; Cai et al., 2018). Several of these approaches such as TransE (Bordes et al., 2013), TransH (Wang et al., 2014), TransG (Xiao et al., 2016) and KG2E (He et al., 2015), to name a few, can be grouped into translational distance models that focus on minimizing a distance based function under some constraints or using regularizing factors between entities and relations. More recent approaches extend these translation approaches by embedding the knowledge graphs into more complex spaces such as the hyperbolic space (Balažević et al., 2019b; Kolyvakis et al., 2020) and the hypercomplex space (Zhang et al., 2019; Sun et al., 2019). Another important class of approaches such as RESCAL (Nickel et al., 2011), DistMult (Yang et al., 2015), TuckER (Balažević et al., 2019a), HypER (Balažević et al., 2019c) and HolE (Nickel et al., 2016) focus on various compositional operators for the entities and relations in the knowledge graph. **Graph Convolutional Networks (GCNs):** Graph Convolutional Networks (GCNs) (Kipf & Welling, 2017) generalize convolutional neural network models to graph-structured data sets where each convolution layer in the GCN applies a graph convolution i.e. a spectral filtering of the graph signal (the feature matrix of the graph) via the Graph Fourier Transform. The main reason behind the success of GCNs is that they exploit two key types of information: node feature descriptions $(x_i)$ and node neighborhood structure (captured through the adjacency matrix $\mathcal{A}$ of the graph).

While successful, GCNs cannot directly be applied on multi-relational data/networks and require propositionalization techniques. Consequently, relational GCNs (Schlichtkrull et al., 2018) construct a latent representation of the entities explicitly and a tensor factorization then exploits these representations for the prediction tasks. We take an alternative approach based on a successful SRL approach (Khot et al., 2014; Lao & Cohen, 2010) to develop novel combinations of the entities and their relationships to construct richer latent representations. As we demonstrate empirically, this leads to superior predictive performance. In addition, the use of *relational rules as the observed layer of the GCN makes them more interpretable/explainable than the tensor factorization approach.*

**Graph Structure Learning (GSL):** There has been growing interest in the problem of graph structure learning (GSL) that refers to learning (near) optimal graph structure from data that can then be used in graph neural networks (GNNs) for downstream tasks (Jin et al., 2020; Zhao et al., 2021; Wu et al., 2022; 2023). These representations have proved to be more effective than using raw input graphs with GNNs especially in real-world applications such as traffic forecasting (Zhang et al., 2020) and drug-drug interaction (Park et al., 2020). The issue with most of the GSL methods is that they only work with single edge type graphs. INTEGRATE can fit in the GSL family although since we deal specifically with multi-relational data with statistical relational learning our work can be thought to be orthogonal to the present GSL algorithms. We point the interested readers to Chen & Wu (2022) for a more detailed discussion on GSL within GNNs.

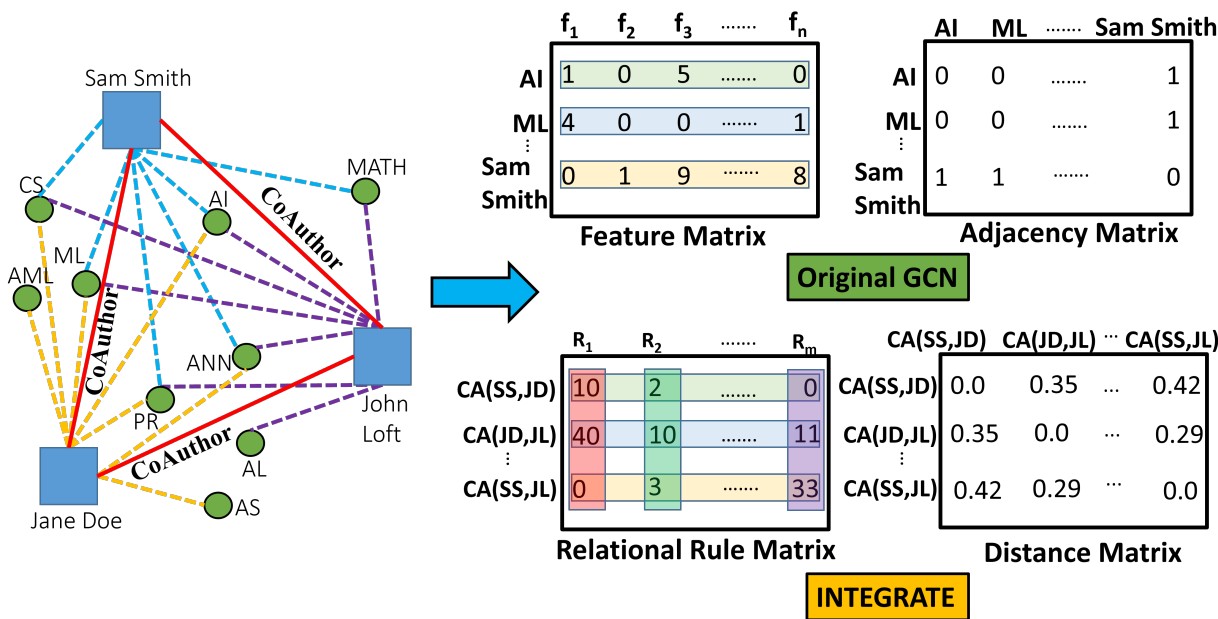

Figure 1: Difference between GCN Kipf & Welling (2017) and INTEGRATE with an example input graph. Here, CA is the CoAuthor relation to be predicted. The relational rule matrix is obtained by counting the number of satisfied groundings of the obtained first-order rules ($R_1$ - $R_m$) wrt the query variables and is a richer representation of the graph structure.

## 3 INTEGRATE (Statistical Relational Learning and GCNs)

Direct application of GCNs cannot fully exploit the inherent structures inside a multi-relational graph. Consequently, they need significant engineering to construct the propositionalized features. Motivated by this, we propose a principled extension to the GCN that models large multi-relational networks faithfully. While a recent work R-GCN (Schlichtkrull et al., 2018) extends GCNs to relational domains, it is still limited to graphs represented as *(subject; predicate; object)* triples and requires multiple adjacency matrices for handling multi-relational data. We propose a novel and a more general approach that is not limited by assumptions about the multi-relationality of the data and can handle general multi-graphs and hypergraphs without loss of information. We can now formally define our model and its components.

**Definition 1 (Secondary Euclidean Graph).** *A secondary Euclidean graph consists of a set of vertices and edges where the vertices correspond to the query variable (which is the relation i.e. the link to be predicted or the node class along with the entities) in relational data and the edges constitute the Euclidean distance between each pair of vertices.*

**Definition 2 (INTEGRATE).** *Given a knowledge base/relational graph $\mathcal{B}$ and a function $\phi : \mathcal{B} \mapsto \mathbb{R}^m$, such that $\phi(\mathcal{B}) = \mathfrak{E} \in \mathbb{R}^m$, INTEGRATE $\mathfrak{G}$ is a graph convolutional network defined over $\mathfrak{E}$ and $Euc(\mathfrak{E})$ i.e. the secondary Euclidean graph.*

**Definition 3 (Relational Rule Matrix).** *A relational rule matrix $\mathcal{X}$ contains the node feature descriptors $x_i \in \mathfrak{E}$ for a Euclidean graph.*

**Definition 4 (Distance Matrix).** *A distance matrix $\mathcal{D}$ contains the Euclidean distances between the node feature descriptors $\in \mathcal{X}$.*

Given a knowledge base $\mathcal{B}$, we first learn a set of first-order rules that captures the relations between the domain predicates. The intuition is that these first-order rules can be viewed as *higher-order features* that connect entities and their attributes. Particularly, when learned for a specific classification task, these features can be both predictive and informative. Given that they are typically conjunctions of relational features (attributes of entities and relationships), they have the added advantage of being interpretable. Our

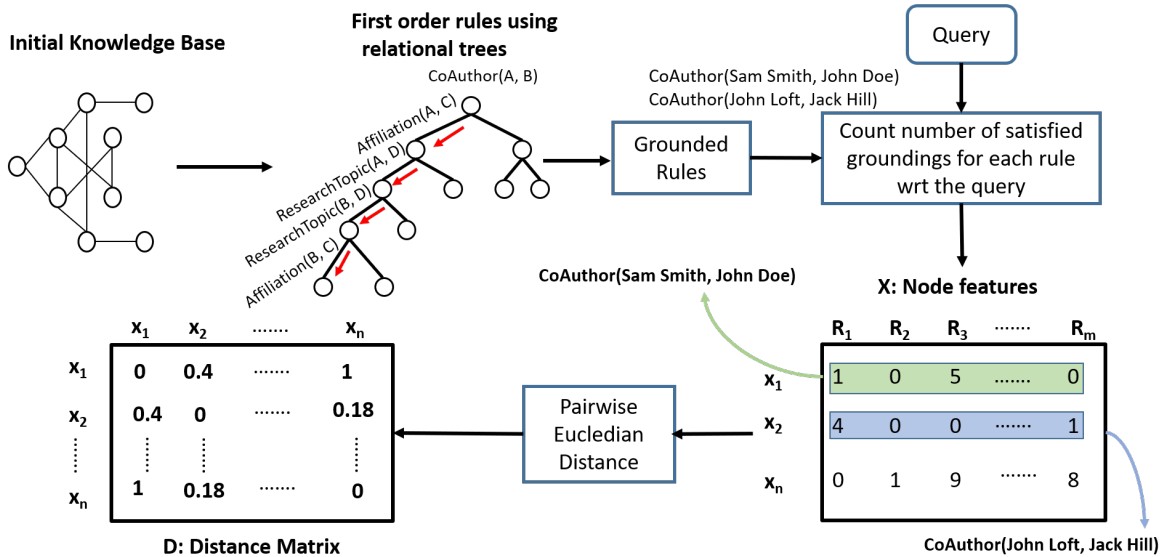

Figure 2: Relational Rule Matrix $\mathcal{X}$ and Distance Matrix $\mathcal{D}$ construction for INTEGRATE. First-order rules are learned from the given knowledge bases which are then grounded and satisfied groundings are counted to form $\mathcal{X}$.

hypothesis, that we verify empirically is that these rules can potentially yield richer latent representations than a relational GCN that simply uses the entity and relationship information.

> *1. One of our key contributions is a two-step process of constructing the link and node classification problems as prediction problems in a secondary Euclidean graph where vertices correspond to target triple rather than individual entities. We learn a relational rule matrix and then build a distance matrix to use for GCN-computations (see Fig. 1).*
>
> *2. Another important difference from typical GCN based methods is that we can handle* n-ary *relations in the data thus allowing for a more general representation.*

Although methods such as HGNN (Feng et al., 2019) and HyperGCN (Yadati et al., 2019) handle n-ary predicates as hypergraphs, they do not take advantage of the relationships between the nodes in the graph and consider only a single type of relation. Also, all of the GCN based method(s) as well as relational embedding methods, require the data to be standardized to the form, $\langle e_1, r, e_2 \rangle$ where $e_1, e_2$ are entities connected by the relation $r$ i.e. handle **only** binary predicates. To handle n-ary predicates they typically decompose the predicates into multiple binary predicates. It is well-known that this process introduces spurious relationships between entities (Kersting & De Raedt, 2008). To present a concrete example, please consider the Carcinogeneisis data set [2] which is also considered in our experimental evaluation. Some of the n-ary relations present in the data set are of the form $sbond1(drug, atom_1, atom_2)$ which signifies that the 2 atoms in a drug have a single bond between them. For example, $sbond1(d1, d1_1, d1_7)$ denotes that the 1st and 7th atom of the drug have a single bond. Other examples are $sbond2(drug, atom_1, atom_2)$ and $sbond3(drug, atom_1, atom_2)$ which signify double and triple bonds between two drug atoms respectively. For classical GCN based methods that cannot handle n-ary relations, the requirement would be to break down these relations into binary relations. Thus, the n-ary relation $sbond1(drug, atom_1, atom_2)$ will result in $sbond1(drug, atom_1)$ and $sbond1(drug, atom_2)$ which are spurious i.e. $sbond1(d1, d1_1, d1_7)$ will result in $sbond1(d1, d1_1)$ and $sbond1(d1, d1_7)$. Another way to handle the n-ary relations might be to create new relations in the original data set only for this single relation, for example, $single_bond(atom_1, atom_2)$ and $drug_atom(drug, atom)$. Similarly $sbond2()$ and $sbond3()$ will result in more spurious or extra relations which

---

[2]https://relational.fit.cvut.cz/dataset/Carcinogenesis

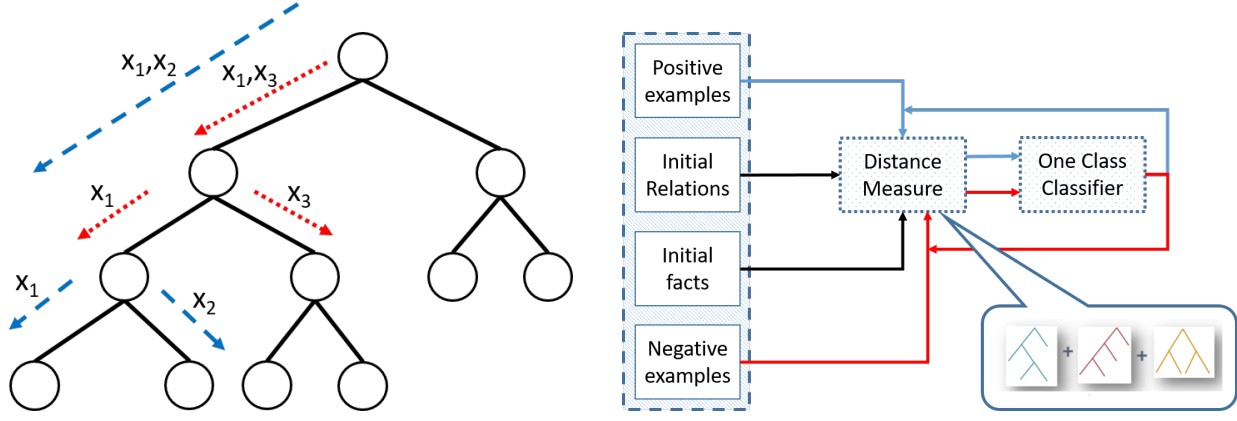

(a) An illustration of Least common ancestor.

(b) Learning the rules/relational features using relational distance. Each left branch of the learned tree represents a relational feature.

Figure 3: Learning the relational examples with the rule learner.

will lead to unnecessary complexity. We can handle n-ary predicates/relations naturally since the underlying *inductive learner uses first-order logic representations.*

### 3.1   Embedding Original Graph to $\mathbb{R}^m$: Creating a Euclidean Graph

We now outline the required steps to embed the original graph to a Euclidean space $\mathbb{R}^n$ thereby creating a secondary Euclidean graph. The nodes of the Euclidean graph consists of the target triple with the node features forming the relational rule matrix $\mathcal{X}$ and the edges connecting the nodes are the Euclidean distances thus forming the distance matrix $\mathcal{D}$. It is clear from def. 1-4 that we just need $\mathcal{X}$ and $\mathcal{D}$ to represent a secondary Euclidean graph and Fig. 2 shows their construction.

**Step 1: Rule Learning using Density Estimation**: Inspired by the success of learning only from positive examples in relational domains (Khot et al., 2014), we learn first-order rules using relational density estimation (which forms $\phi$ in Def. 2) and learn from **both the positive and negative examples separately**. The intuition behind using a density estimation method is that *learning first-order rules for positive and sampled negative examples independently can result in better utilization of the search space thereby (potentially) learning more discriminative features.* Fig. 4 shows an example of learning such discriminative features for a "Co-Author" data set. The density estimation approach uses a tree-based distance measure that iteratively introduces newer features (as short rules) that covers more positive examples.

Thus, we construct a relational graph manifold, by treating relational examples as nodes and connect ones that are close or similar to each other in the neighborhood. The similarity can be measured by learning a tree-based distance between relational examples and is inversely proportional to the depth $d$ of least common ancestor (LCA) of pair of examples, say $x_1, x_2$,

$$d(x_1, x_2) = \begin{cases} 0, & \mathsf{LCA}(x_1, x_2) \text{ is leaf;} \\ e^{-\lambda \cdot \mathsf{depth}(\mathsf{LCA}(x_1, x_2))}, & \text{otherwise,} \end{cases} \tag{1}$$

where $\lambda > 0$ ensures that distance decreases (i.e., similarity increases) as the depth increases.

Fig. 3(a) shows examples $x_1 \equiv$ `advisedBy(Tom, Mary)` and $x_2 \equiv$ `AdvisedBy(Tom, John)`; they both follow the same path down the tree before diverging at a node at depth 2. Now, consider $x_1$ and $x_3 \equiv$ `AdvisedBy(Ada, Dan)`. In this case, we have that the least common ancestor is at depth 1. Since the distance measure is inversely related to depth of the least common ancestor, we have that $x_1$ and $x_2$ are closer together than $x_1$ and $x_3$. We use TILDE (Blockeel & De Raedt, 1998), a first-order tree learner to learn the individual trees. Typically, more than one tree is learned (say, via functional gradient boosting), and the one-class classifier is a weighted combination of these trees. Then, the overall distance function is simply the weighted combination

of the individual tree-level distances: $D(x_1, x_2) = \sum_i \beta_i d_i(x_1, x_2)$ where $\beta_i$ is the weight of the $i^{th}$ tree and $\sum_i \beta_i = 1, \beta_i \geq 0$. The non-parametric function $D(\cdot, \cdot)$ is a relational distance measure learned on the data. The distance function can then be used to compute the density estimate for a new relational example $z$ as a weighted combination of the distance of $z$ from all training examples $x_j$, $E(z \notin \texttt{class}) = \sum_j \alpha_j D(x_j, z)$, where $\alpha_j$ is the weight of the labeled example $x_j$ and $\sum \alpha_j = 1, \alpha_j \geq 0$. Note that expectation above is for $z \notin class$, since the likelihood of class membership of $z$ is inversely proportional to its distance from the training examples describing that class.

We learn a tree-based distance iteratively to introduce new relational features that perform one-class classification. The left-most path in each relational tree is a conjunction of predicates, that is, a clause, which can then be used as a relational feature. The splitting criteria is the squared error over the examples and the goal is to minimize squared error in each node as follows:

$$
\begin{aligned}
&\min \sum_{y \in \mathbf{x_r}} \left[ I(z) - E(z \notin \texttt{class}) - \Sigma_{j:x_j \in \mathbf{x}_l} \alpha_j \beta_i d_i(x_j, z) \right]^2 \\
&+ \sum_{y \in \mathbf{x}_l} \left[ I(z) - E(z \notin \texttt{class}) - \Sigma_{j:x_j \in \mathbf{x}_r} \alpha_j \beta_i d_i(x_j, z) \right]^2
\end{aligned}
\tag{2}
$$

$I(z)$ is the indicator function and returns 1 if $z$ is an unlabeled example or 0 otherwise. Also, $x_l$ and $x_r$ are the examples that take the left and right branch respectively. A greedy search approach is employed for tree learning and since the only parameter is the number of trees that increases as more data is obtained, it thereby provides a *non-parametric* approach for learning these relational trees. Since there is a necessity to learn 2 different set of weights – $\alpha$ and $\beta$, where $\alpha$ is the weight of the example and $\beta$ is the weight of the tree since while calculating the $E(z \notin \texttt{class})$ i.e. $P(z \notin \texttt{class})$ we need to combine distances in two levels, tree level and instance level. These weights are learnt iteratively by minimizing the squared loss function:

$$
\texttt{L} = \sum_{y \in \mathbf{x}} [I(z \notin \texttt{class}) - E(z \notin \texttt{class})]^2
\tag{3}
$$

The different gradients with respect to $\alpha$ and $\beta$ can be calculated as:

$$
\begin{aligned}
\frac{\partial \texttt{L}}{\partial \alpha} &= \frac{\partial \texttt{L}}{\partial \alpha_j} \sum_z [I(z) - E(z \notin \texttt{class})] \\
&= \frac{\partial}{\partial \alpha_j} \sum_z [I(z) - \Sigma_i \alpha_j \beta_i d_i(x_j, z)]^2 \\
&= 2 \sum_z [I(z) - \Sigma_i \alpha_j \beta_i d_i(x_j, z)] \times -\Sigma_i \beta_i d_i(x_j, z)
\end{aligned}
$$

$$
\boxed{\frac{\partial \texttt{L}}{\partial \alpha} = -2 \sum_z [I(z) - E(z \notin \texttt{class})] \Sigma_i \beta_i d_i(x_j, z)}
\tag{4}
$$

$$
\begin{aligned}
\frac{\partial \texttt{L}}{\partial \beta} &= \frac{\partial \texttt{L}}{\partial \beta_i} \sum_z [I(z) - E(z \notin \texttt{class})] \\
&= \frac{\partial}{\partial \beta_i} \sum_z [I(z) - \Sigma_j \alpha_j \beta_i d_i(x_j, z)]^2 \\
&= 2 \sum_z [I(z) - \Sigma_i \alpha_j \beta_i d_i(x_j, z)] \times -\Sigma_j \alpha_j d_i(x_j, z)
\end{aligned}
$$

$$
\boxed{\frac{\partial \texttt{L}}{\partial \beta} = -2 \sum_z [I(z) - E(z \notin \texttt{class})] \Sigma_j \alpha_j d_i(x_j, z)}
\tag{5}
$$

In the tree level combination, we calculate the LCA based distance between all the labeled examples $(x_1, x_2 \ldots x_t)$ and the unlabeled example $z$ *in every learned relational tree* which are then combined to yield

a combined distance $D$ between each example and the unlabeled example. After calculating the distance between each each example with the unlabeled example, a second level of combination is performed to yield the probability of the unlabeled example to belong to a certain class. Fig. 3(b) shows the overall process of learning the relational trees iteratively thereby constructing the rules to be used as features.

We now present some example first order rules learned by the relational one-class classification method for three representative data sets (drug-drug interactions, ICML Co-author and Carcinogenesis) with the last data set being n-ary in nature. The first two rules for each data set are learnt from the positive examples and the next two are learnt for negative examples.

***Data set: Drug-Drug Interactions***

+ Interacts($d_1$, $d_2$) $\implies$ TransporterSubstrate($d_1$, $tr_1$) $\land$ TransporterSubstrate($d_2$, $tr_1$) $\land$ EnzymeInhibitor($d_1$, $e_1$) $\land$ EnzymeInhibitor($d_2$, $e_1$)

+ Interacts($d_1$, $d_2$) $\implies$ EnzymeInducer($d_1$, $e_1$) $\land$ EnzymeSubstrate($d_2$, $e_1$) $\land$ EnzymeInducer($d_2$, $e_2$) $\land$ EnzymeInducer($d_1$, $e_2$)

- Interacts($d_1$, $d_2$) $\implies$ TargetInhibitor($d_1$, $t_1$) $\land$ TargetInhibitor($d_2$, $t_2$) $\land$ TransporterSubstrate($d_1$, $tr_1$)

- Interacts($d_1$, $d_2$) $\implies$ TargetAgonist($d_1$, $t_1$) $\land$ TargetAgonist($d_2$, $t_2$) $\land$ TransporterInducer($d_1$, $tr_1$) $\land$ TransporterInducer($d_2$, $tr_2$)

***Data set: ICML CoAuthor***

+ CoAuthor($p_1$, $p_2$) $\implies$ Affiliation($p_1$, $a_1$) $\land$ Affiliation($p_2$, $a_1$) $\land$ ResearchTopic($p_1$, $topic_1$) $\land$ ResearchTopic($p_2$, $topic_1$)

+ CoAuthor($p_1$, $p_2$) $\implies$ ResearchTopic($p_2$, "Mathematical_Optimization") $\land$ ResearchTopic($p_1$, "Pattern_Recognition") $\land$ ResearchTopic($p_1$,$topic_1$) $\land$ ResearchTopic($p_2$,$topic_1$)

- CoAuthor($p_1$, $p_2$) $\implies$ ResearchTopic($p_1$, "Pattern_Recognition") $\land$ ResearchTopic($p_2$, "Mathematical_Optimization")

- CoAuthor($p_1$, $p_2$) $\implies$ Affiliation($p_1$, "University_of_California_Berkeley") $\land$ Affiliation($p_2$, "Simons_Institute")

***Data set: Caricogenesis***

+ Carcino($d$) $\implies$ drugAtom($d$,$a_1$) $\land$ sbond7($d$,$a_1$,$a_2$) $\land$ sbond1($d$,$a_2$,$a_3$) $\land$ sbond2($d$,$a_3$,$a_4$) $\land$ sbond1($d$,$a_4$,$a_5$) $\land$ sbond1($d$,$a_5$,__)

+ Carcino($d$) $\implies$ drugAtom($d$,$a_1$) $\land$ sbond7($d$, $a_1$, $a_2$) $\land$ sbond1($d$, $a_2$, $a_3$) $\land$ sbond2($d$, $a_3$, __))

- Carcino($d$) $\implies$ drugAtom($d$, $a_1$), sbond2($d$, $a_1$, $a_2$), sbond1($d$, $a_1$, __), sbond1($d$, $a_2$, __))

- Carcino($d$) $\implies$ drugAtom($d$, $a_1$), sbond7($d$, $a_1$, __)

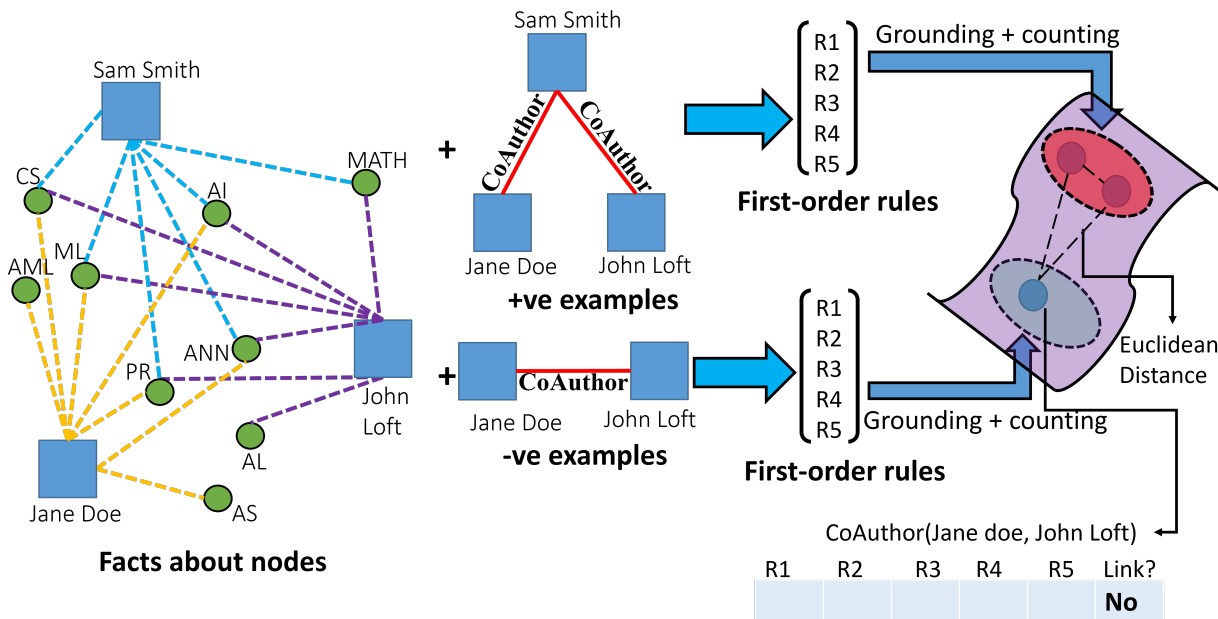

Figure 4: Learning secondary Euclidean graph (nodes) for ICML data set. Learning the +ve and -ve rules and thus features separately result in more discriminative secondary graph nodes with the +ve nodes closer to each other and distant from the -ve node.

After the rule learning there might be an argument about the effect of number of rules on the quality of the learned features. We would like to point out that the number of rules depends on ILP learner which first selects an example from the set of all examples and then finds a rule that best covers the examples. Best covering is the most general clause that covers minimum number of positive examples while excluding a large number of negative examples. Ideal coverage means all positive examples and no negative examples. In practice, this can lead to overfitting and thus we aim to maximize the difference in number of positive and negative examples. Thus, it is important to point out that the number of rules do not matter but rather the quality of rules matter.

**Step 2: Relational Rule Matrix and Distance Matrix Construction:** The learned first-order rules are then grounded to obtain all the instantiations of these rules. The counts of each feature, i.e., the count of the number of times a target example (the coauthor relation between the target entities) is satisfied in every first-order rule is obtained which forms our relational rule matrix $\mathcal{X}$. In spirit, this is similar to MLNs (Richardson & Domingos, 2006) that counts the instances to obtain a marginal distribution. Instead of using the counts to compute marginals, we use them in the matrices. For example, the learned first-order rule from true instances

$$\texttt{CoAuthor}(\texttt{person}_1, \texttt{person}_2) \Leftarrow \texttt{Affiliation}(\texttt{person}_1, \texttt{university}_1)$$
$$\wedge \texttt{Affiliation}(\texttt{person}_2, \texttt{university}_1) \wedge \texttt{ResearchTopic}(\texttt{person}_1, \texttt{topic}_1)$$
$$\wedge \texttt{ResearchTopic}(\texttt{person}_2, \texttt{topic}_1).$$

implies that if two persons have the same affiliation and their research interests lie in same topics, then they are likely to coauthor. Suppose the given target entities are $person_1 = $ "Jane Doe" (JD) and $person_2 = $ "Sam Smith"(SS). The partially grounded first-order rule can then be written as

$$\texttt{CoAuthor}(\texttt{JD}, \texttt{SS}) \Leftarrow \texttt{Affiliation}(\texttt{JD}, \texttt{university}_1)$$
$$\wedge \texttt{Affiliation}(\texttt{SS}, \texttt{university}_1) \wedge \texttt{ResearchTopic}(\texttt{JD}, \texttt{topic}_1)$$
$$\wedge \texttt{ResearchTopic}(\texttt{SS}, \texttt{topic}_1).$$

Then substitutions for all the other entities within the first-order rule are performed and checked whether the substituted first-order rule is satisfied in the groundings. For example, the substitu-

tion $\boldsymbol{\theta} = \{\langle university_1, topic_1 \rangle / \langle UCB, Artificial\ Intelligence \rangle\}$ is satisfied but the substitution $\boldsymbol{\theta} = \{\langle university_1, topic_1 \rangle / \langle UCB, Computer\ Networks \rangle\}$ is not satisfied. Since there can be multiple values taken by $topic_1$ that can satisfy the first-order rule, the count of all such satisfied groundings becomes a feature value for the target query `CoAuthor(Jane Doe, Sam Smith)`. Thus using this satisfiability count we obtain a feature set $\mathcal{X}$ of size $n \times k$ where n is number of target queries and k is number of first-order rules that represent the node features.

In order to obtain the distance matrix $\mathcal{D}$ a pairwise Euclidean distance of all the node feature descriptors i.e. the counts $x_i \in \mathcal{X}$ is computed.

## 3.2 Euclidean Graph GCN

The original GCN formulation (Kipf & Welling, 2017) requires an adjacency matrix $\mathcal{A}$ to perform the layer-wise propagation. Instead of building the adjacency matrix from the relation triples, we use the computed geometric distance matrix $\mathcal{D}$, which is a richer structure (Rouvray & Balaban, 1979; Brouwer & Haemers, 2011), and use it as an approximation to the adjacency matrix for the GCN. To obtain this approximation, we perform the following steps:

**[1]:** A threshold, t, is set as the average of all the distances (since the distance matrix is symmetric, the average is calculated from the upper-right part).

**[2]:** $\forall d_{ij} \in \mathcal{D}$, new distances are computed as $\hat{d}_{ij} = d_{ij}/t$ and $\hat{d}_{ij} > 1$ is set as 1: a far-away case.

**[3]:** Since the higher values in $\mathcal{D}$ represent nodes that are far as opposed to the $\mathcal{A}$ where the higher values i.e. 1 represents the nodes adjacent to each other, the distance between nodes is subtracted from 1 i.e. $\hat{d}_{ij} = 1 - \hat{d}_{ij}$. This is similar to $\mathcal{A}$ with $\hat{d}_{ij} = 1$ representing that two nodes are connected and $\hat{d}_{ij} = 0$ representing that two nodes are not connected with the only difference being the presence of values $0 < \hat{d}_{ij} < 1$ that denote the closeness of two nodes.

The above mentioned approximation was done for 2 major reasons: a) implementation purposes and b) we have a distance matrix which we believe contains richer information of examples; the original GCN takes adjacency matrix to represent structural distances (neighbors, edge connections) among examples. In our graph, the node is not an entity but a triplet w.r.t the query variable. To make it easily understandable and intuitive to work with GCN, we rescale the distance matrix $\mathcal{D}$ to $[0,1]^N * N$ (N = number of nodes). In simpler terms, we do not consider the connection between nodes as 0 (no connection) or 1 (an edge) but we put a weight on each edge, which is a less explored direction in GCNs. To calculate the distance matrix, we use rules as each dimension which are more informative when compared to standard embedding methods.

For INTEGRATE $\mathfrak{G}$ with M layers, the layer wise propagation rule for the layer $l \in$ M can be written as,

$$f(H^{(l)}, \mathcal{D}) = \sigma(\mathcal{D}H^{(l)}W^{(l)}) \tag{6}$$

where $H^{(0)}$ is the input layer i.e. the relational rule matrix $\mathcal{X}$ with $H^{(1)} \ldots H^{(M-1)}$ being the hidden layers. Since we replace $\mathcal{A}$ with $\mathcal{D}$ before the symmetric normalization and addition of self loops, these operations are now performed on $\mathcal{D}$. The updated propagation rule is,

$$f(H^{(l)}, \mathcal{D}) = \sigma(\hat{\mathcal{N}}^{\frac{-1}{2}} \hat{\mathcal{D}} \hat{\mathcal{N}}^{\frac{-1}{2}} H^{(l)} W^{(l)}) \tag{7}$$

such that $\hat{\mathcal{D}} = \mathcal{D} + \mathcal{I}$ where $\mathcal{I}$ is the identity matrix and $\hat{\mathcal{N}} \in \mathbb{R}$ is the diagonal weighted node degree matrix of $\hat{\mathcal{D}}$. In summary, we learn first-order rules from separate densities independently, in the process constructing a secondary graph consisting of query variable as nodes. These learned rules are then grounded resulting in richer representation than simple node features. For obtaining distance between target triples to define adjacency, we use pairwise Euclidean distance. We present our rigorous empirical evaluations next.

## 3.3 Computational Cost

In terms of the **computational cost**, grounding is roughly polynomial in size of the database (and can be reduced by sampling as mentioned above). Counting is exponential in the number of entities and can be

Table 1: Properties of data sets. Note that all node classification data sets except *WebKB* have **n-ary predicates**.

| Data Set | # Rels | # Facts | #+ve Egs | #-ve Egs | # Rules | # Nodes | # Edges |
|----------|--------|---------|----------|----------|---------|---------|---------|
| ICML'18 | 4 | 1395 | 155 | 6498 | 7 | 6653 | 21429036 |
| *ICLR* | 4 | 4730 | 990 | 10000 | 7 | 10990 | 40558762 |
| *DDI* | 14 | 1774 | 2832 | 3188 | 25 | 18060 | 80952471 |
| Carcino | 8 | 54890 | 182 | 258 | 9 | 440 | 45999 |
| *PPMI* | 38 | 314144 | 378 | 812 | 18 | 1190 | 549774 |
| *CiteSeer* | 17 | 119635 | 7504 | 7504 | 7 | 15008 | 54799196 |
| *WebKB* | 5 | 1354 | 153 | 593 | 7 | 746 | 213594 |

reduced by approximate counting. The quadratic amounts of Euclidean distance computations do exist but they can be circumvented by the fact that we only need to compute the upper half of the matrix and can be done parallelly with the help of triangulation (Angeletti et al., 2019).

## 4 Experimental Evaluation

We consider 7 staristical relational AI **data sets** – the first 3 for link prediction and the last 4 for node classification (Tab. 1). *ICML'18* consists of papers from ICML 2018, *ICLR* consists of papers from ICLR (2013-2019) and the prediction task is whether two people are coauthors for both data sets. Both of these data sets are extracted from the Microsoft Academic Graph (MAG) (Sinha et al., 2015). *DDI* is a drug-drug interaction data set (Dhami et al., 2018) and the goal is to predict whether two drugs interact. *Carcino* is a biomedical data set of the structures of chemical compound and the task is to predict if they are carcinogenic. *PPMI* is a study (Marek et al., 2011) designed to identify bio-markers that impact Parkinson's and the task is to predict if a patient has Parkinson's (Dhami et al., 2017). *CiteSeer* is a relational data set of citations (Poon & Domingos, 2007) and the task is to predict the author of a citation. *WebKB* consists of web pages and hyperlinks from 4 CS departments (Craven et al., 1998) and the task is to predict if someone is a faculty. A limitation of our work is that we cannot handle multiple query variables without joint learning where one could consider every relation as the query variable in different rule learning steps to obtain embeddings w.r.t all relations and use them for the knowledge base completion.

We first learn first-order logic rules using relational density estimation (Khot et al., 2014) from positive examples. The number of rules learned each for positive and negative examples is shown in Tab. 1. A secondary Euclidean graph is then constructed with its properties i.e. the number of nodes and edges also shown in Tab. 1. Note that the constructed Euclidean graphs can be very dense in nature. We can circumvent this either by constructing a minimum spanning tree (Loukas, 2020) of the obtained graph or by obtaining its topological minor (which is another graph) (Pilipczuk & Siebertz, 2017) that can then be used with a GCN. We do like to note that both problems are non trivial to solve and thus we will include these are left for immediate future work. The relational rule matrix $\mathcal{X}$ and the distance matrix $\mathcal{D}$ are then obtained from the secondary Euclidean graph.

In INTEGRATE, with the use of first-order logic, node classification problem can be formulated similarly to link predication. For example, suppose a graph consists of 3 nodes $n_1$, $n_2$ and $n_3$, then the potential target relation can be created as: link($n_1$,$n_2$), link($n_1$,$n_3$) and link($n_2$,$n_3$). Analogously, for the node classification problem, suppose we have binary classes, say 1 and 0, we can create potential target relations as class($n_1$,1/0), class($n_2$,1/0) and class($n_3$,1/0). Hence, for node classification, the target is a relation between an object and the class label whereas in link prediction the target is a relation is between two objects.

We aim to answer the following questions through our experimental evaluation:

**(Q1)** How does INTEGRATE perform on data sets that have few relational examples?

**(Q2)** Is learning a secondary graph structure useful?

**(Q3)** How well does our method handle n-ary predicates?

**(Q4)** Can the combination of SRL with deep models such as GCN result in better predictive models?

**(Q5)** How does rule learning from relational density estimation compare with other rule learning methods?

**(Q6)** What is the effect of different distance measures on the performance of INTEGRATE?

**(Q7)** How sensitive is INTEGRATE to the choice of parameters?

### 4.1 Baselines

**Link Prediction:** We compare INTEGRATE, to 15 embedding baselines in 3 categories.

*1. Rule learning (STARAI-based) methods*: **Handwritten rules** (Niepert, 2016): uses Gaifman locality principle Gaifman (1982) to enumerate all *hand-written first-order rules* within the neighborhood of the target/query variables. After obtaining the counts for the satisfied grounded handwritten rules logistic regression is used for prediction. **Neural-LP** (Yang et al., 2017): learns first-order rules by extending the probabilistic differentiable logic system TensorLog (Cohen, 2016). **metapath2vec** (Dong et al., 2017): generates random walks with user defined meta paths and uses a heterogeneous skip-gram model to generate embeddings. **PRAGCN:** makes use of relational random walks (PRA) Lao & Cohen (2010) to learn the first-order rules (Kaur et al., 2019) and obtain the features as described in our method. The learned features are then passed on to a GCN. **Node+LinkFeat** (Toutanova & Chen, 2015) (N+LF): is obtained by running logistic regression (LR) and 2-layer neural network (NN) over the learned propositional features.

*2. Relational embedding methods*: **ComplEx** (Trouillon et al., 2016): proposes a latent factorization approach in multi-relational graphs. We use the ComplEx implementation in the AmpliGraph python library [3]. **ConvE** (Dettmers et al., 2018): uses convolutions over embeddings and fully connected layers to model interactions between input entities and relationships. We use ConvE from AmpliGraph python library. **SimplE** (Kazemi & Poole, 2018): adapts the concept of Canonical Polyadic decomposition and learns two dependent embeddings for each entity and relation to obtain a similarity score for each triple. We use the tensorflow implementation [4]. **ReInceptionE** (Xie et al., 2020): uses a relation-aware networks with joint local-global structural information. **ExpressivE** (Pavlović & Sallinger, 2023): embeds pairs of entities as points and relations as hyper-parallelograms in the virtual triple space. We use the PyTorch implementation [5] and test both variants namely, base (ExpressivE-B) and functional (ExpressivE-F). **HousE** (Li et al., 2022): uses two types of Householder transformations: roation and projection to model relation patterns and mapping properties simultaneously thus resulting in more expressive embeddings. We use the PyTorch implementation [6] and test both variants, namely HousE and HousE+.

*3. GCN based methods*: **Relational GCN** (Schlichtkrull et al., 2018): extends GCN to the relational setting. and can handle different weighted edge types i.e. relations. It uses a 2 step message passing technique to learn new node representations which are then fed to a factorization method, DistMult (Yang et al., 2015). We use the tensorflow implementation[7]. **CompGCN** (Vashishth et al., 2020): jointly embeds both nodes and relations in a graph and we use the PyTorch implementation[8]. **NBFNet** (Zhu et al., 2021): is a graph neural network architecture specifically for link prediction using generalized Bellman-Ford algorithm. We use the PyTorch implementation [9]. **SEAL** (Zhang & Chen, 2018): extracts a local subgraph around each target link thereby learning the best heuristic required for link prediction automatically. We use the PyTorch implementation [10].

**Node Classification:** We compare against 5 relational embedding baselines (covering all 3 categories) and 2 state-of-the-art SRL methods[11]: **MLN-Boost** (Khot et al., 2011) and **RDN-Boost** (Natarajan et al., 2012). For non-SRL methods, we convert the n-ary predicates to $\binom{n}{2}$ binary predicates.

---

[3]https://github.com/Accenture/AmpliGraph

[4]https://github.com/Mehran-k/SimplE

[5]https://github.com/AleksVap/ExpressivE

[6]https://github.com/rui9812/HousE

[7]https://github.com/MichSchli/RelationPrediction

[8]https://github.com/malllabiisc/CompGCN

[9]https://github.com/DeepGraphLearning/NBFNet

[10]https://github.com/muhanzhang/SEAL/tree/master/Python

[11]https://github.com/starling-lab/BoostSRL

| Data | Methods | Recall | Precision | F1 | AUC-PR |
|---|---|---|---|---|---|
| | Handwritten | 0.10 | 0.16 | 0.174 | 0.127 |
| | Neural-LP$_3$ | **0.927** | 0.024 | 0.047 | 0.267 |
| | Neural-LP$_{10}$ | 0.891 | 0.035 | 0.069 | 0.143 |
| | metapath2vec | 0.836 | 0.209 | 0.335 | 0.286 |
| | PRAGCN | 0.0 | 0.0 | 0.0 | 0.512 |
| | ComplEx | 0.85 | 0.013 | 0.03 | 0.04 |
| | ConvE | 0.636 | 0.01 | 0.02 | 0.015 |
| *ICML'18* | SimplE | **0.927** | 0.012 | 0.023 | 0.128 |
| | ReInceptionE | 0.855 | 0.014 | 0.028 | 0.142 |
| | ExpressiveE-B | 0.655 | 0.011 | 0.021 | 0.016 |
| | ExpressiveE-F | 0.709 | 0.011 | 0.022 | 0.018 |
| | HousE | 0.964 | 0.015 | 0.031 | 0.536 |
| | HousE+ | 0.945 | 0.015 | 0.03 | 0.561 |
| | N+LF (LR) | 0.379 | **1.0** | 0.549 | 0.396 |
| | N+LF (NN) | 0.338 | **1.0** | 0.559 | 0.409 |
| | R-GCN | 0.636 | 0.07 | 0.13 | 0.13 |
| | CompGCN | 0.727 | 0.022 | 0.044 | 0.185 |
| | NBFNet | 1.0 | 0.03 | 0.058 | 0.757 |
| | SEAL | 0.879 | 0.112 | 0.199 | 0.776 |
| | **INTEGRATE** | 0.389 | **1.0** | **0.561** | **0.556** |
| | Handwritten | 0.564 | 0.795 | 0.66 | 0.488 |
| | Neural-LP$_3$ | 0.939 | 0.308 | 0.463 | 0.421 |
| | Neural-LP$_{10}$ | **0.987** | 0.275 | 0.429 | 0.453 |
| | metapath2vec | 0.828 | 0.338 | 0.480 | 0.641 |
| | PRAGCN | 0.0 | 0.0 | 0.0 | 0.544 |
| | ComplEx | 0.269 | 0.032 | 0.057 | 0.105 |
| | ConvE | 0.677 | 0.037 | 0.069 | 0.054 |
| *ICLR* | SimplE | 0.973 | 0.054 | 0.102 | 0.535 |
| | ReInceptionE | 0.764 | 0.039 | 0.074 | 0.075 |
| | ExpressiveE-B | 1.0 | 0.058 | 0.11 | 0.96 |
| | ExpressiveE-F | 1.0 | 0.061 | 0.115 | 0.951 |
| | HousE | 1.0 | 0.058 | 0.11 | 0.767 |
| | HousE+ | 1.0 | 0.058 | 0.109 | 0.771 |
| | N+LF (LR) | 0.977 | **1.0** | **0.988** | **0.981** |
| | N+LF (NN) | 0.338 | **1.0** | 0.559 | 0.409 |
| | R-GCN | 0.667 | 0.783 | 0.720 | 0.763 |
| | CompGCN | 0.906 | 0.719 | 0.802 | 0.912 |
| | NBFNet | 0.997 | 0.47 | 0.639 | 0.986 |
| | SEAL | 0.963 | 0.518 | 0.674 | 0.915 |
| | **INTEGRATE** | 0.594 | **1.0** | 0.745 | **0.972** |
| | Handwritten | 0.469 | 0.707 | 0.564 | 0.581 |
| | Neural-LP$_3$ | 0.727 | 0.336 | 0.459 | 0.368 |
| | Neural-LP$_{10}$ | 0.779 | 0.338 | 0.472 | 0.403 |
| | metapath2vec | 0.782 | 0.652 | 0.711 | 0.707 |
| | PRAGCN | 0.427 | 0.700 | 0.531 | 0.695 |
| | ComplEx | 0.832 | 0.492 | 0.618 | 0.705 |
| | ConvE | 0.931 | 0.384 | 0.544 | 0.678 |
| *DDI* | SimplE | 0.992 | 0.288 | 0.446 | 0.503 |
| | ReInceptionE | 0.987 | 0.364 | 0.532 | 0.834 |
| | ExpressiveE-B | 0.985 | 0.383 | 0.552 | 0.876 |
| | ExpressiveE-F | 0.992 | 0.387 | 0.557 | 0.912 |
| | HousE | 0.96 | 0.334 | 0.496 | 0.67 |
| | HousE+ | 0.992 | 0.359 | 0.528 | 0.836 |
| | N+LF (LR) | 0.682 | 0.924 | 0.785 | 0.781 |
| | N+LF (NN) | 0.715 | 0.948 | 0.815 | 0.833 |
| | R-GCN | 0.571 | **1.0** | 0.727 | 0.922 |
| | CompGCN | 0.882 | 0.552 | 0.679 | 0.826 |
| | NBFNet | 0.965 | 0.451 | 0.615 | 0.869 |
| | SEAL | 0.948 | 0.429 | 0.591 | 0.853 |
| | **INTEGRATE** | **0.998** | 0.986 | **0.992** | **0.998** |

Table 2: Link prediction.

| Data | Methods | Recall | Precision | F1 | AUC-PR |
|---|---|---|---|---|---|
| | Neural-LP$_3$ | 0.182 | 0.099 | 0.128 | 0.128 |
| | Neural-LP$_{10}$ | 0.327 | 0.149 | 0.205 | 0.160 |
| | metapath2vec | 0.473 | 0.356 | 0.406 | 0.359 |
| | PRAGCN | 0.0 | 0.0 | 0.0 | 0.5 |
| *Carcino* | R-GCN | 0.259 | 0.789 | 0.390 | 0.573 |
| | N+LF (LR) | 0.529 | **0.973** | 0.686 | 0.734 |
| | N+LF (NN) | 0.550 | 0.957 | 0.698 | 0.537 |
| | MLNB | 0.390 | 0.302 | 0.340 | 0.296 |
| | RDNB | 0.451 | 0.188 | 0.265 | 0.190 |
| | **INTEGRATE** | **0.660** | 0.971 | **0.786** | **0.926** |
| | Neural-LP$_3$ | 0.0 | 0.0 | 0.0 | 0.562 |
| | Neural-LP$_{10}$ | 0.722 | 0.254 | 0.376 | 0.279 |
| | metapath2vec | 0.704 | 0.604 | 0.651 | 0.786 |
| | PRAGCN | 0.287 | 0.829 | 0.426 | 0.618 |
| *PPMI* | R-GCN | 0.712 | 0.771 | 0.740 | 0.729 |
| | N+LF (LR) | 0.354 | **1.0** | 0.523 | 0.568 |
| | N+LF (NN) | 0.342 | **1.0** | 0.509 | 0.342 |
| | MLNB | 0.684 | 0.972 | 0.803 | **0.967** |
| | RDNB | **0.816** | 0.886 | **0.849** | 0.950 |
| | **INTEGRATE** | 0.436 | **1.0** | 0.607 | 0.798 |
| | Neural-LP$_3$ | 0.0 | 0.0 | 0.0 | 0.615 |
| | Neural-LP$_{10}$ | 0.500 | 0.231 | 0.316 | 0.443 |
| | metapath2vec | 0.905 | 0.923 | 0.914 | 0.976 |
| | PRAGCN | **1.0** | 0.490 | 0.657 | 0.745 |
| *CiteSeer* | R-GCN | 0.971 | 0.958 | **0.964** | **0.991** |
| | N+LF (LR) | 0.787 | 0.681 | 0.730 | 0.641 |
| | N+LF (NN) | 0.823 | 0.888 | 0.854 | 0.782 |
| | MLNB | 0.942 | **0.975** | 0.958 | 0.979 |
| | RDNB | 0.948 | 0.942 | 0.947 | 0.979 |
| | **INTEGRATE** | 0.780 | 0.681 | 0.727 | 0.818 |
| | Neural-LP$_3$ | 0.0 | 0.0 | 0.0 | 0.533 |
| | Neural-LP$_{10}$ | 0.362 | 0.082 | 0.133 | 0.086 |
| | metapath2vec | 0.426 | 0.204 | 0.276 | 0.192 |
| | PRAGCN | 0.317 | 0.929 | 0.473 | 0.698 |
| *WebKB* | R-GCN | 0.200 | 0.225 | 0.212 | 0.251 |
| | N+LF (LR) | 0.375 | 0.913 | 0.532 | 0.484 |
| | N+LF (NN) | 0.403 | **1.0** | 0.574 | 0.403 |
| | MLNB | **1.0** | **1.0** | **1.0** | **1.0** |
| | RDNB | **1.0** | **1.0** | **1.0** | **1.0** |
| | **INTEGRATE** | 0.220 | **1.0** | 0.360 | 0.611 |

Table 3: Node classification.

## 4.2 Results

For INTEGRATE, we use a GCN with 2 hidden layers each with dimension = 16 with a drop out layer between the 2 graph convolutional layers. To introduce non-linearity, we use *ReLU* between input and hidden layers and to score queries, we use *log_softmax* function. The examples for training, validation and testing are randomly sampled without replacement. For neural embedding baselines, since they are trained on true relations, the positive examples are randomly split to $\langle 60\%, 10\%, 30\% \rangle$ in training, validation and testing respectively. To obtain the different metrics for the neural embedding baseline, the scores for each pair of nodes in the test examples were thresholded by the average of the obtained scores. If the score between pair of nodes $\geq$ average score the link is predicted to be true. We run our experiments on a GPU with 8 GeForce GTX 1080 Ti cards.

**(Q1. Smaller data sets)** Tab. 2 shows the result of link prediction task. Our method outperforms all the baselines significantly in 2 of the 3 data sets with the difference being significant in the smaller data set *ICML'18* and is comparable in the *ICLR* data set. Note that although the recall is high for the neural embedding baselines, the corresponding F1 score and AUC-PR are low which implies that the **baseline relational embedding methods have a high rate of false positives**. This clearly demonstrates that

Table 4: Effect of change in type of GCN

| Data | Method | Recall | Precision | F1 Score | AUC-PR |
|------|--------|--------|-----------|----------|--------|
| *ICML'18* | heat-diffusion + GCN | 0.348 | **1.0** | 0.516 | 0.544 |
| | ppr + GCN | 0.351 | **1.0** | 0.520 | 0.538 |
| | INTEGRATE | **0.369** | **1.0** | **0.539** | **0.692** |
| *Carcino* | heat-diffusion + GCN | 0.604 | 0.967 | 0.743 | 0.896 |
| | ppr + GCN | 0.609 | 0.921 | 0.725 | 0.880 |
| | INTEGRATE | **0.660** | **0.971** | **0.786** | **0.926** |

INTEGRATE is significantly better than the strong baselines for the link prediction task. Tab. 3 shows the result of node classification task and our method outperforms the SRL and GCN baselines. Specifically, our method is significantly better in the smaller data sets of *Carcino* and *PPMI*. This answers **Q1** affirmatively.

**(Q2. Secondary graph/distance matrix impact)** The main advantage of our method is learning a secondary graph structure where both link prediction and node classification tasks become simple prediction tasks in this new graph. As can be seen from the results for link prediction, a simple discriminative machine learning algorithm (logistic regression), used on top of the learned features (N+LF) performs better than the other baselines including GCN-based baselines. In case of node classification, the results are comparable.

To show the importance of using a distance matrix, we compare our method with Graph Attention Networks (GATs) (Veličković et al., 2018) which uses the adjacency matrix. Fig. 5 shows the results and it can be seen **that using a distance matrix can be an effective alternative**. This is expected since in the secondary structure, as the nodes show the query, there is no particular notion of connection between the nodes. Fig. 6 show an extended comparison of INTEGRATE with variations of GAT using the distance matrix (GAT ADJ), converting distance matrix to binary adjacency matrix (GAT ADJ), and finally using PRA features along with the adjacency and distance matrices (PRAGAT ADJ and PRAGAT DIS). The results show that the performance of PRAGAT is lower which is due to the fact that PRA features may hurt the performance due to its quality. The sparser binary adjacency matrix generated from the converted distance matrix brings non-negative effect on approaches with PRAGAT and GAT. This answers **Q2** affirmatively.

**(Q3. n-ary predicates)** Note that all node classification data sets except *WebKB* have n-ary predicates and thus GCN methods cannot handle them naturally. Since we use a logic learner to learn first-order rules and use resolution for grounding, we can easily handle n-ary predicates. Our results in Tab. 3 show that we can handle n-ary predicates to produce richer representation of underlying features to answer **Q3**. R-GCN demonstrates the best performance in *CiteSeer* data since, due to the large size of the data set, introduction of spurious relations do not have an adverse impact. The comparison between our method and the N+LF baseline which uses the exact same features is notable. Our method is more stable than N+LF confirming the richness of learned abstract features.

**(Q4. SRL + GCN)** Our results show that using SRL models (relational density estimation in our case) as the underlying feature learner which are then fed to a neural model, GCN, gives us a powerful hybrid model that can be used seamlessly with relational data. Using a SRL model as the initial layer of a neural model results in learning richer initial features set used by the neural model. This initial feature set can take advantage of underlying graph structure faithfully and thus, in accordance, **leads to the neural model learning far richer abstract features which in turn leads to better predictive performance**. Our evaluations on both tasks support this as our method significantly outperforms GCN baselines in 6/7 (especially in the smaller) domains. We also replace the vanilla GCN with more powerful diffusion based GCNs (Klicpera et al., 2019) and INTEGRATE still outperforms (Tab. 4) thus answering **Q4** affirmatively.

**(Q5. Effective rule learning)** To answer **Q5**, we use 4 different rule learning methods: handwritten rules (Gaifman), NeuralLP (rule length 3 & 10), metapath2vec and PRA. Tabs. 2 and 3 clearly demonstrate that using our *density estimation method significantly outperforms all rule learning method across all domains*. Comparing PRAGCN and INTEGRATE is especially interesting since this shows that rule learning method plays a crucial role in learning richer features, especially in the imbalanced domains, where relational density estimation is demonstrably beneficial since both methods share the underlying GCN. The difference in

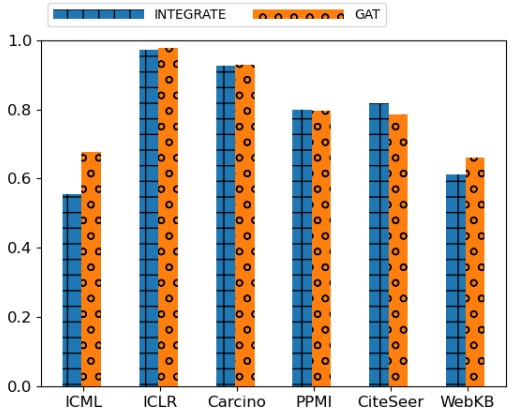

Figure 5: Comparison (AUC-PR) with GATs showing the importance of $\mathcal{D}$. DDI does not run using GAT.

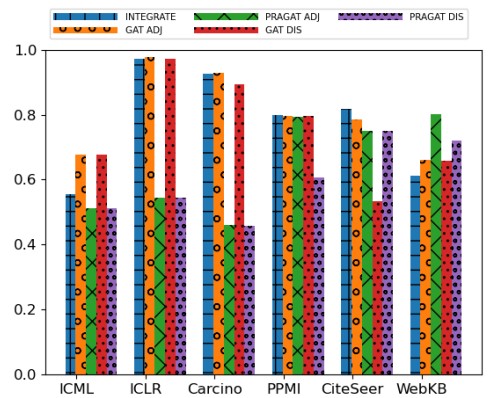

Figure 6: AUC-PR comparison with variations of GAT with PRA features, distance matrix and distance matrix as adjacency matrix.

Table 5: Effect of size of hidden layers.

| Data | Size | Recall | Precision | F1 Score | AUC-PR |
|------|------|--------|-----------|----------|--------|
| | 32 | 0.369 | 1.0 | 0.539 | 0.692 |
| *ICML'18* | 64 | 0.369 | 1.0 | 0.539 | 0.692 |
| | 128 | 0.369 | 1.0 | 0.539 | 0.692 |
| | 32 | 0.989 | 0.998 | 0.993 | 0.998 |
| *DDI* | 64 | 0.989 | 0.998 | 0.993 | 0.998 |
| | 128 | 0.989 | 0.998 | 0.993 | 0.998 |

Table 6: Effect of number of hidden layers.

| Data | # | Recall | Precision | F1 Score | AUC-PR |
|------|---|--------|-----------|----------|--------|
| | 3 | 0.369 | 1.0 | 0.539 | 0.692 |
| *ICML'18* | 4 | 0.369 | 1.0 | 0.539 | 0.692 |
| | 5 | 0.369 | 1.0 | 0.539 | 0.692 |
| | 3 | 0.989 | 0.998 | 0.994 | 0.999 |
| *DDI* | 4 | 1.0 | 0.997 | 0.998 | 0.999 |
| | 5 | 0.999 | 0.991 | 0.995 | 0.997 |

performance of PRAGCN and INTEGRATE is **significantly high** in the highly imbalanced domains *ICML'18* and *ICLR* where the features learned by the PRA method result in all examples being classified as negative. In node classification experiments, PRA features classify *all the examples* in *Carcino* and *CiteSeer* as positive. Note that **the features learned by using PRA are biased towards a single (the larger) density across all domains.** This answers **Q5**.

**(Q6. Effect of distance measures)** Fig. 7 presents the effect of 2 other distance measures, Manhattan ($L_1$) and Chebyshev ($L_\infty$), in addition to Euclidean ($L_2$) on the performance of INTEGRATE on the DDI data set. Since Euclidean is the shortest distance between nodes, it performs the best.This answers **Q6**.

**(Q7. Effect of parameter choices)** To answer **Q7**, we change the size of the hidden layers in the GCN as well as the number of hidden layers and test our method on *ICML'18* and *DDI* data sets. Tabs. 5 and 6 show that change of these parameters have none or very minuscule effect on the overall results. This answers **Q7** and also shows that the ***learned features by themselves are quite expressive thus removing the need for a more complex GCN.***

## 5  Discussion

The main motivation behind using relational density estimation to create a two-step learning process for GCNs is that learning first-order rules for positive and sampled negative examples independently can result in better utilization of the search space resulting in creating richer features for training. Another motivation is that the relational density estimation can result in better discriminative features in both the relational as well as the Euclidean space. A TSNE proejction for the drug-drug interaction (DDI) data set can be seen in Fig. 8 that shows that learning the rules from different densities separately does . Our approach *is inspired by manifold learning methods such as Laplacian eigenmaps that construct a graph representation of the data manifold by treating training examples as nodes and connecting them to other similar nodes in their neighborhood.* We hypothesize that the following reasons result in our model learning a richer set of features:

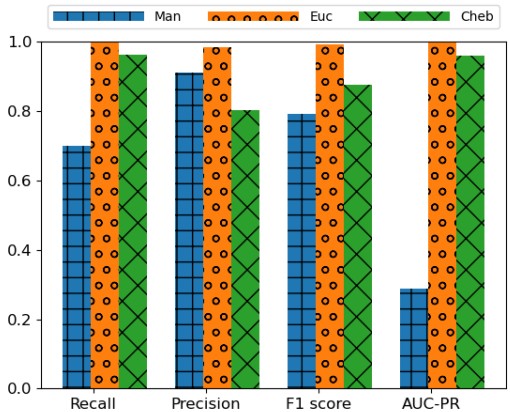

Figure 7: Effect of the choice of distance measure on the link prediction results for the DDI data.

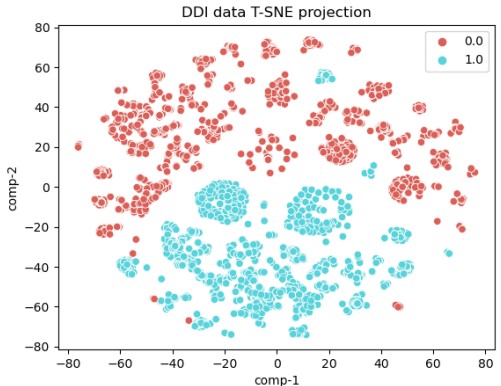

Figure 8: A TSNE plot for the DDI data. Learning from +ve and -ve data separately in relational domain results in discriminative features when projected to Euclidean space.

**1.** The features constructed by our method encapsulate the information about entities as well as the relationship between the entities in the relational space while GCN-based methods reconstruct the relationships using a real valued vector via a decoder. The real valued vector inevitably leads to a loss of information while the first-order features capture the graph structure faithfully thereby leading to richer features.

**2.** GCN captures the node features as well as the neighborhood of the nodes using the adjacency matrix to perform the predictive tasks. In our model we have 2 levels of neighborhood information aggregation – first, while learning the first-order rules using the relational distance and second in the underlying GCN model which results in richer feature learning which results in a stronger model.

**3.** While constructing the relational rule matrix $\mathcal{X}$, the grounding process takes into account the current node and its immediate neighbors, thus effectively creating another round of neighborhood aggregation.

**4.** The tree-based distance in density estimation is a learnable, non-parametric, relational metric that can be used to characterize the adjacency/distance of relational examples effectively.

**5.** Graph convolution is the SOTA approach effectively capturing information stored in graph structure and node features. Using it on a secondary graph means essentially taking advantage of the graph structure twice.

There might be a concern regarding the generalizability of INTEGRATE when compared to GCNs. There are two ways to look at the generalization question. When considering the point of view of changes in structure of the graph, the generalizability of our method is limited when compared to classical GCN and GCN based methods. But when viewed from the traditional lens of statistical relational learning generalization where the number of instances or objects in the domain can change, our method is relatively more general.

## 6    Conclusion

We presented the first GCN method that can learn from multi-relational data utilizing the different densities separately. Our method does not make assumptions on the supervision/arity of predicates and automatically constructs rules that allow for a rich latent representation. We significantly outperform the recently successful methods on KB completion tasks across multiple data sets. Allowing for joint learning and inference over multiple types of relations is an important future direction. Using more classical rule learning techniques

such as Quinlan (1990); Muggleton (1995); Srinivasan (2001) is another interesting direction. Scalability is a major issue for various SRL systems but we can circumvent this by adopting approximation techniques. The counting operation to create the relational rule matrix, along with the grounding operation, presents the biggest overhead. In future work, we plan to integrate sampling and approximate counting methods (Das et al., 2019) that will reduce the learning time considerably without sacrificing performance. Finally, learning in the presence of hidden/latent data and rich human domain knowledge is essential for deploying SRL methods in real tasks.

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
