# OpenReview forum: "INTEGRATE: Distance based Graph Convolutional Networks for Statistical Relational Learning"
_TMLR — Rejected by TMLR_

### Review · Reviewer_JrT8 · 2023-04-25

**Summary Of Contributions:**

This paper studies graph neural network approaches for learning representations of entities and relations in knowledge graphs. The proposed approach is subject to much empirical investigation that demonstrates the effectiveness of using the proposed distance-matrix based approach and euclidean graph input to GCNs.

**Audience:**

Yes

**Broader Impact Concerns:**

None.

**Claims And Evidence:**

No

**Requested Changes:**

Please see Weaknesses for high level changes.

### Writing

* Abstract: "inevitability" -- not clear to me this is the right word or at least what is meant here is unclear to me.
* Abstract: "semantic structure of the network is not exploited" -- not clear from previous part of the abstract why this is so.
* Introduction: Claim in first sentence should be backed by citations.
* Introduction "semantic structure of the network is not exploited" claim should have citations. Though I don't believe this to be entirely true, e.g. [Schlichtkrull et al, (2018)'s R-GCN](https://arxiv.org/abs/1703.06103). I think the claim should be about multi-relational structure? as is in the second section?
* I think the INTEGRATE name is great :) but maybe there is a way to make the acronym fit it more closely?
* Definition 2. sorry, it is unclear to me what $\mathbb{B}$ is? should it be $\mathcal{B}$ ?
* Definition 4. sorry, while i understand the concept of a distance matrix, I don't think I understand this definition. The $Euc(\textfrak{E})$ is a member of the distance matrix?
* With the emphasis on non-binary relations, it would be nice to have the example figure reflect this
* "A greedy search approach..." implies that the greediness makes it non-parametric? where as in fact it is the use of the tree structure of instances that makes it non-parametric?
* Not clear we need gradient derivation in main paper.
* What are the distinctions between Khot et al., (2014) and the Step 1: Rule Learning using Density Estimation?
* Examples of rules could probably have been better arranged in a figure rather than body?
* please remove place holder appendix

**Strengths And Weaknesses:**

### Strengths

* The paper presents a detailed and empirically investigated approach for modeling entities and relations in knowledge graphs
* The approach consists of an interesting mix between non-parametric & deep GCN-like models.

### Weaknesses

Methodological
* I find the presentation of the contributions of the methodology a bit confusing. In general I have the following doubts:
* First, I see the separation of the representation learning + GCN part a bit confusing. I think its not that the adjacency matrix is not the starting point, its just about the objective that is used to learn the representations that go into the later part of the network. This is happening for all such models as I understand, it is just that a particular objective is proposed here? I'd love the authors to help clarify my understanding here. I understand you emphasize this as a contribution, but the novelty and distinction from other GNN models is not clear to me.
* Second, I find the presentation around n-ary relations a bit confusing. I think it would be clearer to present more precisely with more examples about (1) why previous work does not support these (2) the ingredients and evaluation of the proposed approach in n-ary relations.
* I think that the claims such as "first GCN method that can learn from multi-relational data utilizing the different densities
separately" were a bit surprising to me. After reading the paper, I was not entirely convinced that previous work could not have also met the criteria proposed, especially with all the variations of the proposed approach that combine it with other works, e.g. GAT.

Empirical
* n-ary predicates - It is not obvious to me that this experiment answers Q3. "handle n-ary predicates". It seems to answer whether using n-ary relations improves node classification. I would have expected this to be about predicting n-ary relations.
* Q1 - Not sure I understand -- are these binary relation prediction experiments? Sorry, I am still a bit confused about the setup of this.
* Why is Figure 6 not included in the Q5 / Q6 discussion?
* Table 5 & 6 - These show little variation. Perhaps means there is more to explore to find where the variation is?
* My personal view - I think that the presentation would be improved (in this readers opinion) without such strong language about an empirical question being answered. Personally, I think a strong presentation presents the hypothesis and findings and tells the reader why this is the most likely hypothesis to answer the question. Perhaps even saying why other hypotheses to explain the results are less probably given the observed evidence.

---

> ### Author Response · Authors · 2023-05-21
> **Response to the review (1/2)**
>
> We thank the reviewer for the thoughtful review and would like to point out that the concrete suggestions did help us improve the manuscript to a large extent. We now address the raised concerns pointwise below:
>
> **Methodological**
>
> **Q: Separation of the representation learning + GCN part**
>
> A: We would like to refer the reviewer to 1st para of page 2 where we explicitly mention the motivation behing seperating the representation learning and GCN part. To put it again "Since the two different steps of learning the relational rules and training the GCN employ the same set of positive examples, a richer representation of the combination of the attributes, entities and their relations is obtained. While previous methods used the features as the observed layer, INTEGRATE uses the rules as the observed layer. This has the added advantage of the latent layer being richer – it combines the instantiations of first-order rules themselves allowing for a richer representation" As mentioned in the paper, motivation of the model design is that we want to learn a richer set of node representations by learning a secondary graph." Most of the GCN models use raw features from the graph. Even if there are methods that try and learn optimal features, they do not handle multi-relational data. Also the density estimation part is essentially used to learn the first-order rules where we learn from **positive and negative examples separately** which can result in the learned Euclidean graph being more discriminative. We would also like to point the reviewer to the discussion section in the paper.
>
> **Q: Claim of "First GCN method that can learn from multi-relational data utilizing the different densities separately"**
>
> A: To the best of our knowledge ours is the first work to do the one-class learning of features to construct a secondary Euclidean graph and then use this newly learned graph in GCNs.
>
> **Q: n-ary predictaes**
>
> A: while everything can be decomposed as binary predicates, there has been several work in logic and probabilistic logic (see for instance the work of Kersting et al on Bayesian logic programs that we had cited) where it is shown that spurious combinations can be introduced by this decomposition. This can induce unnecessary bias and lead to significantly worse results as we have mentioned explicitly in the paper.
>
> To present a concrete example from the used data sets as requested by the reviewer we consider the Carcino data set. Some of the n-ary relations present in the data set are of the form sbond1(drug,atom1_id,atom2_id) [for example, sbond1(d1,d1_1,d1_7)] which signifies that the 2 atoms in a drug have a single bond between them [in our example the 1st and 7th atom of the drug have a single bond]. Other examples are sbond2(drug,atom1_id,atom2_id) and sbond3(drug,atom1_id,atom2_id) which signify double and triple bonds between two drug atoms respectively.
>
> If our method was not able to handle n-ary relations, the requirement would be to break down these relations into binary relations. Thus, for the n-ary relation sbond1(drug,atom1_id,atom2_id) will result in sbond1(drug,atom1_id) and sbond1(drug,atom2_id) which are spurious i.e. sbond1(d1,d1_1,d1_7) -> sbond1(d1,d1_1) and sbond1(d1,d1_7).
>
> Another way to handle this might be to create a couple of new relations in the original data set only for this single relation, for example, single_bond(atom1_id,atom2_id), drug_atom(drug,atom_id). Similarly sbond2() and sbond3() will result in more spurious or extra relations which will lead to unnecessary complexity. We have included this discussion in the paper in page 5.

---

> > ### Author Response · Authors · 2023-05-21
> > **Response to the review (2/2)**
> >
> > **Empirical**
> >
> > **Q: Experiment does not answer Q3. "handle n-ary predicates"**
> >
> > A: There seems to be some misunderstanding and we would like to politely disagree with the reviewer here. The Carcino data set is n-ary by nature and since we have a first-order logic tree learning in the representation learning part we can handle such n-ary predicates naturally. There is no notion of improving node classification by adding n-ary predicates in our work, although this can be a very interesting avenue for future research.
> >
> >
> > **Q: Smaller data sets**
> >
> > A: By smaller data sets we mean the relational data sets with less number of relations and facts. GCNs generally require comparatively larger number of examples to train and we claim here that due to the richer set of feature representations learnt due to density estimation, our model can even work with data sets with smaller set of examples. We have made this clearer in the paper.
> >
> > **Q: Fig 6 not included in answering Q5 and 6**
> >
> > A: The reviewer is right that Fig 6 could have been potentially used for answering the said questions as well. But we feel that it fits the best for Q2, as the experiment is designed to show the difference between using adjacency matrix and disstance matrix and that the sparser nature of distance matrix does not have much ill effect on the performance of the GCN.
> >
> > **Q: Little variation in Tab. 5 and 6**
> >
> > A: This is actually a positive result for our model as mentioned in the answer to Q 7 on Pg 15. Of course, finding a variation can provide more information about the underlying features learnt but we tried several configurations and did not find much variation at all in the downstream task performance thereby showing that the learned feature representation is expressive enough.
> >
> > **Q: Strong language about an empirical question being answered**
> >
> > A: We completely understand the reviewer sentiment. We would however like to argue that the language is not very strong in our opinion. There is a classic way of writing research questions that a given method is trying to tackle and then trying to answer the questions based on the perfomed experimentations. When we say we can affirmitavely answer the questons, it is of course based on the performed experimentation and the achieved results (which is hopefully implicitly understood by the reader). We hope that the reviewer does not hold this against us as this is a matter of personal taste as the reviewer also correctly points out.
> >
> > **Writing**
> >
> > We have made all the minor changes that the reviewer had pointed out.
> >
> > 1. replaced "inevitability" by "necessity"
> >
> > 2. replaced "semantic structure of the network is not exploited" by "inherent semantic structure of the network is not exploited since only the node features are taken into account by these set of methods"
> >
> > 3. added citations in 1st line of introduction.
> >
> > 4. replaced "semantic structure of the network is not exploited" with "multi-relational structure of the network is not exploited" in both abstract and introduction.
> >
> > 5. fitting the acronym INTEGRATE: Apart from it fitting the name relatIonal deNsity disTance basEd GRAph convoluTional nEtworks,we chose this specific acronym since our work "integrtaes" the areas of graph neural networks and Statistical relational learning :)
> >
> > 6. Def. 2: correct. changed to $\mathfrak{B}$
> >
> > 7. Def. 4: we understand the confusion. We removed the Euc($\mathfrak{E}$) $\in \mathcal{D}$.
> >
> > 8. n-ary in figure: we understand the reviewer's point of view but we feel that the figures depict the underlying process of INTEGRATE nicely. Also the n-ary predicates are not a focus but a nice property of our method.
> >
> > 9. non-parametric nature: The non-parametric nature comes from the fact that the only parameter is the number of trees that increases as more data is obtained. We have made the required change in the paper. Thank you pointing it out.
> >
> > 10. gradient derivation is a nice depiction of learning separate set if weights for examples and trees and thus we kept it in the text.
> >
> > 11. For learning the trees, we use the same loss as presented in Khot et al., AAAI 2014
> >
> > 12. We tried to arrange the rules in a figure but the results were not optimal. We therefore would have to keep the rules in the text and hope it is fine with the reviewer.
> >
> > 13. removed the appendix.

---

### Review · Reviewer_jNdR · 2023-04-29

**Summary Of Contributions:**


This paper presents a new framework, named INTEGRATE, for relational GCNs for statistical relational learning. Instead of using the original graph structure, it creates a secondary Euclidean graph by learning rules from one-class data using a relational density estimation technique. The graphs vertices correspond to the target triples and the edges correspond to the Euclidean distances between the target triples. Different from traditional relational GCNs, INTEGRATE can handle n-ary relations. Experiments show that INTEGRATE outperform other baselines on KB completion tasks across multiple datasets.


**Audience:**

Yes

**Claims And Evidence:**

Yes

**Requested Changes:**

Please refer to the above weakness points.

**Strengths And Weaknesses:**

Strengths:

*Combination of GCNs and SRL and creating a new graph from rule learning are interesting and new.
* The proposed framework can handle n-ary relations.
* Good experimental performance and through analysis of the advantages of INTEGRATE compared to other baselines. I also like the showcase of learned rules.




Weaknesses:

* Writing. I am not an expert in statistical relational learning or rule learning. It was a pain to read this paper. But I think it is not only because I am not familiar with some terminologies or techniques, but also due to bad organization of the paper, many inaccurate expressions as well as lack of explanations. For example:
	* how is the node classification formulated in the framework? What is the corresponding vertex in this task?
	* the vertices are the target triples, but in Definition 1 the authors says the vertices correspond to query variable, what does “which is the relation i.e. the link to be predicted or the node class along with the entities” mean? How can the “the node class along with the entities” be a triple?
	* In Section 3.1, what is a class of a relational example “z”? Any examples?
	* In step 1 of Section 3.1, how is the initial tree (e.g. the one in Figure2) constructed?

* After we have the relational rule matrix, we can directly use an MLP to do the prediction. What is the benefit of learning another distance matrix and then use GCN? I think this comparison should be added in experiments.

* In traditional GCNs, once they have the trained embeddings they can infer all missing relations or node classes. Unlike them, the proposed approach has to use the target triples as vertices. This limits the generalizability of the model.

---

> ### Author Response · Authors · 2023-05-21
> **Response to the review**
>
> We thank the reviewer for the thoughtful review and would like to point out that the concrete suggestions did help us improve the manuscript to a large extent. We now address the raised concerns pointwise below:
>
> **Q: How is node classification formulated?**
>
> A: In INTEGRATE, with the use of first-order logic, node classification problem can be formulated similarly to link predication. For example, suppose a graph consists of 3 nodes $n_1$, $n_2$ and $n_3$, then the potential target relation can be created as: link($n_1$,$n_2$), link($n_1$,$n_3$) and link($n_2$,$n_3$). Analogously, for the node classification problem, suppose we have binary classes, say 1 and 0, we can create potential target relations as class_relation($n_1$,1/0), class_relation($n_2$,1/0) and class_relation($n_3$,1/0). Hence, for node classification, the target is a relation between an object and the class label whereas in link prediction the target is a relation is between two objects. We have added this discussion in the experimental section of the paper on page 11.
>
> **Q: Meaning of "which is the relation i.e. the link to be predicted or the node class along with the entities"**
>
> A: The statement means that for link prediction the vertex of the graph is either link($entity_1$,$entity_2$) as depicted in Fig.4 i.e. CoAuthor(Jane Doe, John Loft) and for node classification it is the class_relation($entity$,class_label). For example, consider the Carcinogenesis data set where the task is to decide if a given compound is carcinogenic in nature. The vertex of graph will then be Carcino(compound,1/0).
>
> **Q:  What is a class of a relational example “z”?**
>
> A: It is the relation to be predicted which is again the link or the node class. For example the class can be CoAuthor for the ICML data set and Carcino = true or false for the Carcinogenesis data set.
>
> **Q: In step 1 of Section 3.1, how is the initial tree (e.g. the one in Figure2) constructed?**
>
> A: We use TILDE [1], a first-order tree learner to learn the individual trees and we use functional gradient boosting to learn multiple TILDE trees and fit the data. We missed to cite TILDE in the paper and have done so now. Thank you for pointing this out.
>
> **Q: After we have the relational rule matrix, we can directly use an MLP .**
>
> A: The reviewer is right and we already had experiments on this in the original submission under the Node+LinkFeat baseline. The idea to use GCN on the learned features is to have another level of neighborhood aggregation as well as taking advantage of graph structure twice as already mentioned in introduction and discussion section.
>
> **Q: Limited generalizability**
>
> A: We would like to thank the reviewer for bringing this important point up. The reviewer is absolutely correct in noticing that when considering the point of view of changes in structure of the graph, the generalizability of our method is limited when compared to classical GCN and GCN based methods. But we would like to argue that when viewed from the traditional lens of Statistical relational learning genralization where the number of instances or objects in the domain can change, our method is more general than the classic GCN and GCN based methods. Thank you again for bringing this point up. We have added this in the paper in the discussion section.
>
> **References:**
>
> [1] Blockeel, H. and De Raedt, L., 1998. Top-down induction of first-order logical decision trees. Artificial intelligence, 101(1-2), pp.285-297.

---

### Review · Reviewer_L9Ro · 2023-05-08

**Summary Of Contributions:**

This paper studies the problem of learning the latent graph structure for GCNs. The authors propose INTEGRATE, a framework that learns distance-based latent graph by density estimation and applies GNN afterward. The proposed framework can boost GCNs on node classification and link prediction tasks.



**Audience:**

Yes

**Broader Impact Concerns:**

no concern in particular

**Claims And Evidence:**

Yes

**Requested Changes:**

See above

**Strengths And Weaknesses:**

Strong points:
S1. The problem that only true relation triplets are labeled while the false instances are not explicitly identified is vital in the KG community.
S2. Learning the graph structure by a density estimation problem is a novel approach.
S3. The method provides a solution for graph structure generation for KG.
S4. INTEGRATE boosts GCNs on both node classification and link prediction tasks.

Weak points:
W1. The studied problem is an important yet well-studied problem. There have been a ton of works studying graph structure learning, i.e. how to learn an optimal structure for GNN towards downstream tasks. This line of work should be compared or at least discussed.
W2. The proposed method is likely to suffer from scalability issues.
W3. Different from standard KGC datasets e.g. FB15k-237, WN18RR, the authors use three non-standard link prediction datasets. Also, some of the node classification datasets are less-adopted in the GNN community.
W3. The proposed methods are compared against weak baselines. Most methods are uncompetitive and out-of-date (before 2020), more recently proposed methods should be compared, e.g. [HousE], [Buddy], [ExpressivE]. [SEAL] should also be compared. Besides, recent literature on KGC methods is not discussed in the related work.
[SEAL] Link Prediction Based on Graph Neural Networks. In NIPS 2018.
[HousE]: Knowledge Graph Embedding with Householder Parameterization. In ICML 2022.
[ExpressivE]: A Spatio-Functional Embedding For Knowledge Graph Completion. In ICLR 2023.
[Buddy] Graph Neural Networks for Link Prediction with Subgraph Sketching. In ICLR 2023.

---

> ### Author Response · Authors · 2023-05-21
> **Response to the review**
>
> We thank the reviewer for the thoughtful review and would like to point out that the concrete suggestions did help us improve the manuscript to a large extent. We now address the raised concerns pointwise below:
>
> **Q: Graph Structure Learning**
>
> A: Thank you for pointing this out and this is indeed relevant line of work. We have added a new paragraph within related work for the same. We would like to point out that GAT can be thought of as a learning a weighted graph structure as pointed out in [1].
>
> **Q: Scalability**
>
> A: Scalability is indeed a major issue for various SRL systems but we can circumvent this by adopting approximation techniques as we had already mentioned in the conclusion section of the paper. The counting operation to create the relational rule matrix, along with the grounding operation, presents the biggest overhead. In future work, we plan to integrate sampling and approximate counting methods [3] that will reduce the learning time considerably without sacrificing performance. In terms of the computational cost, grounding is roughly polynomial in size of the database (and can be reduced by sampling as mentioned above). Counting is exponential in the number of entities and can be reduced by approximate counting. The quadratic amounts of Euclidean distance computations do exist but they can be circumvented by the fact that we only need to compute the upper half of the matrix and can be done parallelly with the help of triangulation[4].
>
> **Q: Non-standard data sets and weak baselines**
>
> Regarding exclusion of FB15k-237 and WN18RR data, the rules in our method are learned with respect to a fixed predicate (query) as mentioned in the paper. Thus standard relational learning data sets are more natural for evaluation. Our immediate direction is to perform joint learning where one could consider every relation as the query variable in different rule learning steps to obtain embeddings w.r.t all relations and use them for the knowledge base completion tasks (similar to RDNs) (as mentioned in the limitation of our method in page 11, 1st paragraph last 3 lines). Also, the standard datasets are of the form (id entity1-relation id-id entity 2) where a proper FOL rule is difficult to learn since no relations are given explicitly. Also note that since we explicitly wanted to adapt GCNs for statistical relational learning (STARAI) we considered the specific STARAI data sets.
>
> We have now included 3 out of the 4 baselines mentioned by the reviewer, SEAL, ExpressiveE (both base and functional version) and HousE (both base and + version) in **Table 3** for link prediction. We see that INTEGRATE is easily able to outperform these baselines. We could not make the subgraph-sketching code work.
>
> **References**
>
> 1. Chen, Y. and Wu, L., 2022. Graph neural networks: Graph structure learning. Graph Neural Networks: Foundations, Frontiers, and Applications, pp.297-321.
>
> 2. Das, M., Wu, Y., Khot, T., Kersting, K. and Natarajan, S. 2016. Scaling lifted probabilistic inference and learning via graph databases. In Proceedings of the SIAM International Conference on Data Mining.
>
> 3. Angeletti, M., Bonny, J.M. and Koko, J., 2019. Parallel Euclidean distance matrix computation on big datasets.

---

### Decision · Action_Editors · 2023-06-20

**Recommendation:** Reject

**Comment:**

The reviewers found the method interesting especially the creation of another euclidean graph learned from relational density estimation, and that can then be used for other end tasks.  But they also had many comments and questions for the authors. The replies to those questions were appreciated and clarified some concerns, in particular the new experiments and baselines. But during the final discussion, all reviewers agreed that the paper cannot be accepted in this current form.

Some feedback to improve the paper from the reviewers discussion are provided below :

- Positioning with recent GNN approaches that also address the question of structure learning is limited. The added paragraph in the introduction is appreciated but short and does not really position the proposed method with the existing approaches which is important. The paper would also greatly benefit from additional presentation of why previous work's handling multi-relational data is insufficient.
- All reviewers agreed that the paper is very hard to read, especially  for a general ML readership, and that the writing and presentation could be greatly improved. This is also related to the next point.
- The very important discussion and  presentation of n-ary relations could be improved and the questions QX asked in the numerical experiments should be more precise since they can only be answered partially with those experiments.
- It is good that the authors added some more recent baselines. Similarly the comparison with GAT in Figure 5 and 6 is very interesting but  it also leaves the reader with questions about when to use the proposed distance matrix compared to adjacency matrix.
- The presentation of results, to include all methods in each table rather than separate them (e.g., no GAT in Table 2) is somewhat confusing. Perhaps, the separation in performance would have been clearer in datasets, which are more challenging (many of the methods seem to achieve perfect or near perfect precision/recall).
- The reviewers were concerned about the scalability of the method and its generalization (to new graphs). The reply about generalization from the authors both in openreview and added to the paper is interesting but a bit vague and deserves more details (generalization wth respect to what, possibly with examples).

We refer to the reviews for more details.


**Audience:**

The paper address important questions at the interface of GNN and knowledge bases, the contributions and questions raised by this paper would be of definite interest to some individuals in the TMLR audience.

**Claims And Evidence:**

The claims  of the papers in term of contributions seem to be accurate and supported. But reviewers found that the paper is sometimes unclear and lacks some details and clarifications especially on the discussion and answers to the questions asked in the numerical experiments.

**Resubmission Of Major Revision:**

The authors may consider submitting a major revision at a later time.